# Nutritional supplement containing a nuclear fraction of bovine thymus gland increases the circulating levels of spermidine

Natalia Surzenko[1], Ashley Dominique[1], Taleen Hanania[2], Melville Osborne[2], Bassem F. El-Khodor [1]*

1 Nutrition Innovation Center, Standard Process Inc., Kannapolis, North Carolina, United States of America, 2 PsychoGenics, Inc., Paramus, New Jersey, United States of America

* belkhodor@standardprocess.com

## Abstract

Polyamines (PAs), including spermidine, spermine and their precursor, putrescine, are ubiquitous molecules that are vital for a variety of physiological processes. Recently, PAs gained research attention because of their roles in promoting longevity and preventing age-related diseases. Circulating and tissue levels of PAs appear to decline with age, while higher intake of PAs in humans is correlated with better health during aging. Many foods, including plants and offal (organ meats), are good sources of dietary PAs, but are consumed much less in regions with prevailing Western diets. Elevating the circulating levels of PAs through dietary supplementation with PA-rich plant extracts or foods, on the other hand, has proven to be challenging, most likely due to their low bioavailability. In this study, we evaluated the effectiveness of nutritional supplements derived from bovine glandular tissues and/or plant material in elevating blood and tissue levels of spermidine, spermine and putrescine in adult rats. We detected appreciable amounts of PAs in the following materials: 1) spermidine-rich supplement (SRS), containing wheat germ, 2) a cytosolic fraction extract of bovine thymus gland (Thymus Cytosolic Fraction – TCF) and 3) a nuclear fraction extract of bovine thymus gland (Thymus Nuclear Fraction – TNF). We showed that all three PA-containing supplements also contain liposomes, with TNF displaying the largest amounts of liposomal PAs, as well as RNAs, among the tested supplements. We demonstrated that oral administration of SRS, TCF and TNF induce rapid changes in blood PA concentrations. Finally, we showed that TNF supplement is superior to SRS and TCF in elevating the levels of spermidine in the blood, liver and heart following a 28-day supplementation period. Considering the importance of PAs in prevention of age-related disease, supplementation with TNF could be a plausible approach towards the maintenance of proper cellular PA homeostasis during aging.

**Data availability statement:** All relevant data are within the paper and its Supporting Information files.

**Funding:** This study was funded by Standard Process, Inc. The funder provided support in the form of salaries for authors B.F.E.-K., A.D. and N.S., but did not have any additional role in the study design, data collection and analysis, decision to publish, or preparation of the manuscript.

**Competing interests:** N.S., A.D., and B.F.E.-K. are salaried employees of Standard Process Inc.; all other authors declare no conflicts of interest.

## Introduction

Polyamines (PAs), namely spermidine and spermine, including their precursor putrescine, are naturally occurring, positively charged alkylamines that are present in every living cell and are essential for a wide range of cellular physiological processes and functions [1]. PAs are abundant across all kingdoms of life and have attracted the attention of researchers because of their many diverse functional roles in cardio- and neuroprotection, neuromodulation, lifespan extension, and prevention of age-related diseases [1–12]. Multiple lines of research evidence suggest that higher dietary intakes of PAs are associated with lower mortality rates and reduced incidence of diseases in humans [11–17]. In addition, recent studies demonstrated an important relationship between blood PA concentrations, the ratios of individual PAs, and healthy life span [17–19].

PAs have essential and diverse cellular functions, which include structural stabilization of RNA and DNA [20–22], DNA condensation [23], RNA processing [24,25], regulation of gene expression [25,26], protein synthesis and posttranslational modification [27], reducing oxidative stress [28], promoting cell growth and cell cycle progression [29], and possible regulation of autophagy, mitophagy, apoptosis and inflammation [30–34], among others [7,35–39]. Essentially, PAs are vital for cell survival, and their depletion would severely compromise cellular growth and function [40]. As organisms age, however, PA levels in tissues decline, contributing to the disruption of many molecular pathways associated with age-related diseases [41–44].

The pools of PAs in the body are maintained through three distinct routes: 1) endogenous synthesis, 2) production of PAs by the microorganisms residing in the intestinal tract, and 3) dietary intake – the most important and also modifiable source of PAs [10]. Among the dietary sources of PAs are diverse foods of plant and animal origins, many of which are found abundantly in Mediterranean diets that are thought to promote longevity and prevent age-related diseases [14,39,41,45,46]. Among the plant sources, wheat germ is one of the richest in PAs, especially spermidine [47]. High concentrations of PAs are also found in foods of animal origin, such as edible offal (organ meats), including intestine, thymus and liver [14,45,48,49]. Of note, consumption of offal and other foods that are common in Mediterranean diets (i.e., fermented foods, vegetables, fruit), is much lower in the countries with predominant Western diets, which may have an impact on overall population longevity and health during aging [50,51]. PAs that come from either the diet or gut microflora are absorbed intact in the gut lumen, but their concentrations drop from millimolar to low micromolar ranges rapidly after ingestion through poorly understood mechanisms, making PAs "high supply and low utilization" compounds [52,53]. Even though specialized carriers have been proposed to mediate PA transport across intestinal enterocytes, their affinity for PAs may differ based on their location at the apical versus basal membranes of enterocytes, while PAs that are absorbed are quickly distributed across tissues with high proliferative demands, including the cells of the gastrointestinal tract itself.[54–56] In addition, PAs that reach rat tissues 6 hours following exogenous PA administration mediate a 30−300 fold increase in the activity of

spermidine/spermine N-1-acetyltransferase (SSAT), an enzyme involved in the conversion of higher PAs back to putrescine, thereby limiting their availability.[57]

Consistent with this, supplementation with high doses of spermidine, whether in the form of a supplement or in the context of spermidine-rich foods, does not increase its levels in plasma or in saliva [58–60]. Multiple studies in humans and in rodents demonstrated that supplementation with foods rich in PAs, even for as long as 1 year in humans, does not increase blood spermidine levels, while spermine levels can gradually increase [11,53,61]. These studies confirm that the bioavailability of food-derived spermidine is very low. Indeed, concentrations of PAs in the blood depend on the maintenance of PA homeostasis, which is a tightly controlled and highly dynamic process [17,56,62]. Putrescine is a substrate for the synthesis of spermidine, which can be further converted to spermine, while a series of acetylation and oxidation reactions mediate the conversion of the higher PAs back to their precursors (Fig 1) [63]. A set of enzymes and antizymes, whose activity and/or expression are regulated by the abundance of their substrates, putrescine or higher PAs, as well as by many environmental factors, including injuries or pathologic conditions, mediates rapid bi-directional interconversion of each of the PAs into a different type and their import, export, metabolism and catabolism, and controls the changes in PA ratios. Yet, blood PA levels, and PA ratios specifically, can serve as biomarkers of human health status [64–66].

Importantly, PA catabolism increases following traumatic brain injury (TBI) and other types of insults to the central nervous system (CNS), leading to reduced spermidine levels in the brain and in the blood [67–69]. Spermidine supplementation, on the other hand, was shown to exert a neuroprotective effect in the context of TBI, such that the recovery from TBI is enhanced in rodents receiving spermidine prior to and post-injury [69]. Coincidentally, we recently demonstrated a neuroprotective effect of a nutritional supplement containing a nuclear fraction of bovine thymus gland (Thymus – Nuclear Fraction, TNF) by showing that supplementation of rats with TNF prior to and post TBI accelerated their functional

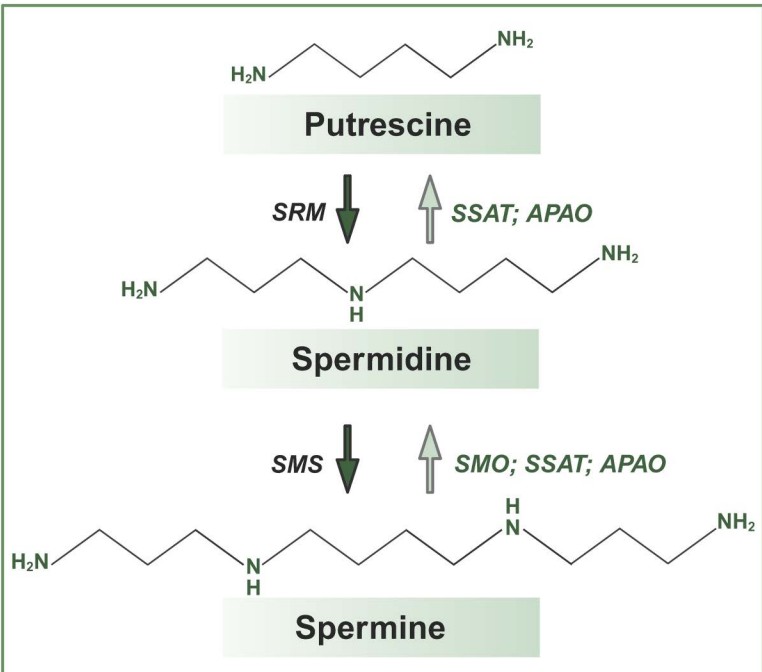

**Fig 1. Pathways of PA synthesis and interconversion.** Spermidine is synthesized from putrescine by spermidine synthase (SRM), and spermine is produced from spermidine by spermine synthase (SMS). Direct conversion of spermine to spermidine is mediated by spermine oxidase (SMO). Additional pathways converting spermine to spermidine, and spermidine to putrescine, require sequential actions of spermine/spermidine N$^1$-acetyltransferase (SSAT) and N$^1$-acetylpolyamine oxidase (APAO) enzymes. Illustration created in Biorender.com.

recovery [70]. Given that thymus is a glandular organ that could potentially be a good dietary source of PAs [45,48,49], we hypothesized that TNF supplement may deliver bioavailable PAs, which would partially explain the observed neuroprotective role of TNF supplementation.

In the current study we investigated whether TNF supplement contains PAs and could change their circulating levels when administered orally. We compared a whole food supplement containing defatted wheat germ, termed spermidine-rich supplement (SRS), with bovine thymus extracts – TNF, thymus cytosolic fraction (TCF) and thymus desiccated (freeze dried) material (TD), as well as nuclear fractions of bovine liver and heart (LNF and HNF), with respect to their PA content and effectiveness in increasing blood and tissue PA levels. We show that SRS contains the highest amount of spermidine among the tested supplements, yet TNF is a supplement that predominantly delivers the liposomal, bioavailable form of spermidine and is capable of elevating its levels in the blood, liver and heart following 28 days of oral supplementation in rats. Together, these results represent the first demonstration that a nutritional supplement containing a nuclear fraction of bovine thymus is a source of bioavailable forms of PAs and can increase the levels of spermidine in the blood and in tissues.

## Materials and methods

### Nutritional supplements

All nutritional supplements evaluated in this study were obtained from Standard Process Inc. (Palmyra, WI, USA) and are commercially available as nutritional supplements. Cytoplasmic and nuclear extracts were prepared from frozen raw bovine glands and dried on-site using Standard Process Inc. proprietary processes. Briefly, the cytoplasmic cellular material was extracted following incubation of the raw homogenized thymus tissue in water, while the nuclear material was isolated using salts following the removal of the cytoplasmic material. No extraction procedures were applied towards the desiccated, freeze-dried thymus material. All test products are in powder form and are suitable for consumption by animals and humans. The following supplements were used: 1) Spermidine-Rich supplement (SRS) containing plant- and bovine tissue-derived materials (a proprietary blend containing bovine liver, organic beet [root], nutritional yeast, defatted wheat germ, rice bran, organic sweet potato, organic carrot, and bovine adrenal gland), 2) bovine thymus cytosolic fraction extract (TCF), 3) bovine thymus nuclear fraction extract (TNF), 4) desiccated bovine thymus (TD), 5) bovine liver nuclear fraction extract (LNF) and 6) bovine heart nuclear fraction extract (HNF).

### PA measurements in nutritional supplement samples

Measurements of individual PA concentrations in nutritional supplements were conducted by Eurofins Scientific (Luxembourg) using high-performance liquid chromatography method EN ISO 19343, essentially as described in Duflos et al. (2019) [71]. Two lots of materials were tested for each supplement.

### Cryogenic Transmission Electron Microscopy (Cryo-TEM)

Cryo-TEM was performed at Creative Biostructure, LLC (Shirley, New York, USA). Solutions were prepared from supplement powders at a concentration of 100 mg/mL, centrifuged, and the supernatant was used in grid preparation. A 5 µl solution aliquot was placed on a thin copper grid that had been glow discharged. For preparation of the grid, the samples were loaded into the freezing chamber at low temperature (0–5°C) under humidity control (100%). After blotting for 3 seconds (s) with filter paper, the specimen was rapidly frozen with cryogen – liquid ethane cooled by liquid nitrogen. The prepared grid was mounted for imaging using a 200 kV FEI Tecnai F20 electron microscope.

### Liposomal particle size distribution analysis using dynamic light scattering (DLS)

DLS analysis was conducted at Creative Biostructure, LLC (Shirley, New York, USA). Ten mg of each supplement powder were resuspended in 0.1 mL of water. After centrifugation, the supernatant was used in particle size analysis following the Zetasizer Nano Series manufacturer's standard protocol (Malvern Panalytical, London, United Kingdom).

## Nanoparticle Tracking Analysis (NTA)

NTA analysis was performed at Creative Biostructure, LLC (Shirley, New York, USA). The NanoSight NS300 instrument (Malvern Panalytical, London, United Kingdom) was calibrated using 100 nm beads and primed with filtered 1X PBS since the nanoparticles are isolated in PBS. The stock dilutions of nanoparticles were prepared in 1X PBS and samples were injected into the instrument using a 1 cc syringe. Individual particles were visualized in real-time and NTA was performed at 25 °C using a 488 nm laser. Data were collected from 11 positions.

## PA detection using HPLC for liposomal encapsulation efficiency

HPLC analysis was performed at Creative Biostructure, LLC (Shirley, New York, USA) essentially as described in Tamim et al (2002) [72]. Briefly, HPLC method consisted of a perchloric acid extraction step, derivatization step using dansyl chloride, separation step using elution gradient of methanol and water, and fluorescence-based detection. HPLC results were calculated using a calibration curve created using the same methodology as used for sample analysis. One gram of each supplement powder was resuspended in 10 mL of water and centrifuged to remove the insoluble precipitate. The supernatant was then centrifuged at 150,000 g for 2 hrs to collect the liposomal nanoparticles. After centrifugation, the liposomal nanoparticles were resuspended in 10 mL water and thoroughly mixed with 0.5% triton X-100, and the liposomes were destroyed by slight shock. The suspension was centrifuged at 12,000 g at 4°C for 10 min, and the supernatant was collected. Amounts of loaded spermidine, spermine and putrescine were determined by HPLC. Encapsulation efficiency was determined using the following formula:

$$\text{Encapsulation efficiency} = (\text{Loaded PA content} / \text{Total PA content}) \times 100$$

S1 Table shows the detailed procedures used for characterization of PA encapsulation. The detection limit of the HPLC assay used was 5 ppm; 1 ppm = 1 µg/mL.

## RNA quantification assay

RNA analysis was performed at Creative Biostructure, LLC (Shirley, New York, USA) following the Quant-iT™ RNA Assay Kit's standard manual (Thermo Fisher Scientific, High Point, NC, USA). RNA standards were prepared by adding 10 µL of each of the *E. coli* rRNA standard in diluted RNA buffer to separate wells in triplicate and mixing. Ten µL of each unknown RNA sample were added to separate wells and mixed well. Two hundred µL of the working solution (diluted Quant-iT™ RNA reagent) were loaded onto each microplate well and fluorescence was measured using a microplate reader (excitation/emission maxima were 644/673 nm) and RNA amounts were determined using the *E. coli* rRNA-based standard curve. The encapsulation efficiency for RNA followed the same method described in the section above.

## DNA quantification assay

DNA quantification was performed at Creative Biostructure, LLC (Shirley, New York, USA) using AlphaLISA Host Cell Residual DNA Detection Kit (Revvity, Waltham, MA, USA). DNA Detection Buffer (1X) was made by adding 10 mL of 10X buffer stock solution and 2 mL of UltraPure BSA to 88 mL of pure water. MIX solutions (2.5X) were prepared by adding 50 µL of 5 mg/mL Anti DNA Acceptor beads and 50 µL of 500 nM Biotinylated Anti DNA to 9.9 mL of 1X DNA Detection Buffer. Streptavidin (SA) Donor beads solution (2X; 20 µg/mL) was prepared by adding 50 µL of 5 mg/mL SA-Donor beads to 12.45 mL of 1X DNA Detection Buffer. For DNA amount quantification, 5 µL of each sample was incubated with 20 µL of 2.5X MIX solution for 1 hour at 23°C, followed by adding 25 µL of 2X SA- Donor beads solution and incubation for 30 min at 23°C. Plates were read using SpectraMax® M5e Multi-Mode Microplate Reader (Molecular Devices, Wokingham, United Kingdom) at 615nm. The encapsulation efficiency for DNA followed the same method described above.

## Animals

All animal studies were conducted at PsychoGenics, Inc. (Paramus, New Jersey, USA); all housing and testing of the rats were conducted in accordance with the Principles of Laboratory Animal Care and the approval of PsychoGenics Inc., Institutional Animal Care and Use Committee in AAALAC-accredited facilities. A total of eighteen male Sprague-Dawley rats, 9 weeks old, fitted with jugular vein catheters (JVCs), weighing 200–250 g at arrival (Envigo, Indianapolis, IN, USA), and 18 male SpragueDawley rats, 9 weeks old, without JVCs, weighing 200–250 g at arrival (Envigo, Indianapolis, IN, USA), were used for the acute and chronic supplementation studies, respectively. Animal catheterization was performed at Envigo (Indianapolis, IN, USA) prior to their arrival. Animals were assigned unique identification numbers (tail marks), were housed in pairs in ventilated cages and were examined, handled, weighed and acclimated for 7 days prior to the initiation of the study to assure adequate health and suitability. 12/12 light/dark cycles, 20–23°C room temperature and a relative humidity of approximately 50% were maintained during the study. Body weights (BWs) were measured 2 times per week throughout the study. Chow (Lab diet 5001, LabDiets, St. Louis, MO, USA) and water were provided ad libitum. Testing was conducted during the light cycle, and rats in the same cage received the same treatment.

## Study 1 – acute dosing

Rats with JVCs were balanced by BW and assigned to one of the 3 treatment groups, with each group consisting of 6 rats. Animals housed in the same cage all received the same treatment. Tested nutritional supplements (raw material powders containing active ingredients) were administered orally (oral gavage, PO) at a dose volume of 10 ml/kg as follows: SRS – 645 mg/kg body weight (BW), TCF – 1276 mg/BW and TNF – 1411 mg/BW. These doses delivered approximately 0.12–0.2 mg/kg BW of spermidine and 0.05–0.13 mg/kg BW of spermine and were calculated based on the varying amounts of PAs and excipients present in each supplement. Blood (300 µl) was collected from JVC at the following timepoints: 0 min (blood drawing prior to dosing, baseline), 30 min, 1 hour, 2 hours, 4 hours and 6 hours after dose administration.[73,74] Blood was collected into lithium heparin coated tubes and flash frozen in liquid nitrogen. Samples were stored at – 80°C. Rats were decapitated using guillotine at the end of the study.

## Study 2 – chronic dosing

Rats were balanced by BW and assigned to one of the 3 treatment groups, with each group consisting of 6 rats. Animals in the same cage all received the same treatment. Nutritional supplements were administered orally, 2 times per day, at a dose volume of 5 ml/kg and a dose of 175 mg/kg BW (350 mg/kg BW per day) for 28 days. Considering an average human weight of 65 kg and a conversion factor of 6.2 between rat and human, the human equivalent dose would be approximately 3.4 grams per day [75]. The approximate daily amounts of spermidine delivered by this dose for each supplement were as follows: SRS – 0.072 mg/kg BW, TCF – 0.034 mg/kg BW, and TNF – 0.050 mg/kg BW. This daily dose given to rats is equivalent to human doses of 0.76, 0.36 and 0.53 mg of spermidine for SRS, TCF and TNF, respectively. For baseline blood collection, blood (300 µl) was collected from tail vein into lithium heparin coated tubes and flash frozen in liquid nitrogen and stored at – 80°C. Animals were not administered tested supplements on the day of terminal tissue collection (Day 29). On Day 29, Rats were decapitated using guilootine, and trunk blood was collected into lithium heparin coated tubes and flash frozen on dry ice. Brains, hearts and livers were weighed, and flash frozen on dry ice. All tissue samples were stored at −80°C.

## PA measurements in blood and tissue samples

Quantitative analysis of a total of 198 blood/tissue samples for the amounts of putrescine, spermidine and spermine was performed by Creative Proteomics (Shirley, NY, USA) using an AB SCIEX API 4000 mass spectrometry instrument (Danaher Corporation, Washington, D.C., USA) connected to Waters Acquity ultra-performance liquid chromatography (UPLC) instrument (Waters Corp, Milford, MA, USA).

### Preparation of blood samples

Fifty μL of each blood sample were mixed with 200 μL of ice-cold methanol (with 0.5% formic acid) in a 2 mL tube and vortexed for 3 min. After 10 min of ultrasound-assisted extraction at 4°C, the mixture was centrifuged at 17,000 g at 4°C for 15 min. The supernatant was transferred to a new tube and filtered through a 0.22 μm membrane filter. The sample was further diluted in pure water (with 0.5% formic acid) and 10 μL of the sample were injected into the UPLC-MS/MS for analysis.

### Preparation of tissue samples

All tissue samples (brain, liver and heart) were lyophilized before weighing. Briefly, 50 mg of each sample and 3 steel beads were placed into a 2.0 mL tube and 1 mL water (with 0.5% formic acid) was added to each tube. The samples were homogenized by a bead beater for 10 min. After 10 min ultrasound-assisted extraction at 4°C, the mixture was centrifuged at 17,000 g and 4°C for 15 min. The supernatant was transferred to a new tube and filtered through a 0.22 μm membrane filter. Each sample was further diluted in pure water (with 0.5% formic acid) and 10 μL of the sample was injected into the UPLC-MS/MS for analysis.

Putrescine, spermidine and spermine standards were dissolved in pure water (with 0.5% formic acid) to achieve 1 mg/mL stocking solutions. The standards were diluted in pure water (with 0.5% formic acid) to get gradient concentrations from 1 ng/mL to 200 ng/mL. Ten μL of each sample were used for the UPLC-MS/MS. Chromatographic condition: Waters Acquity UPLC HSS T3 column (2.1 × 150 mm 1.8 μm) coupled with a VanGuard precolumn (2.1 × 5 mm 1.8 μm). LC parameters: Mobile Phase A – pure water (with 0.5% formic acid); Mobile Phase B – acetonitrile (with 0.1% formic acid). The column temperature was held at 40°C. The sample chamber temperature was held at 10°C. Flow rate was 0.25 mL/min. The elution gradients are shown in S2 Table. Mass Spectrometry parameters: MS condition – ESI positive mode; Ion source temperature – 450°C; IS – 4500 V; CAD – 10 psi; CUR – 40 psi; GS1–45 psi; GS2–45 psi; Q1 and Q3 parameters for measured PAs are shown in S3 Table.

### Statistical analysis

Acute changes in blood PA levels in *Study 1* were analyzed using Repeated Measures ANOVA and Bonferroni/Dunn post-hoc test. Paired Student's t-test was used to compare changes in blood PA levels from Day 0 to day 29, following 28 days of supplementation in *Study 2*. Changes in tissue PA levels were compared using Bonferroni/Dunn post-hoc analysis. Significant changes with p values < 0.05 are marked by an asterisk (*). Graphical data analyses were performed using GraphPad Prism 9 software (La Jolla, CA, USA, RRID: SCR_002798).

## Results

### Varying amounts of PAs are found in nutritional supplements containing plant- and bovine organ tissue-derived materials

To begin investigating whether nutritional supplements that are composed of materials derived from plants/animal tissues contain PAs we evaluated SRS, TCF, TNF, LNF, HNF materials, and the freeze-dried, desiccated thymus material (TD), for concentrations of putrescine, spermidine, spermine, histamine and cadaverine. We found that the SRS, which consists of both plant- and bovine tissue-derived materials, contained the highest levels of spermidine among the tested supplements (206.73 mg/kg) (Fig 2). The cytosolic and the nuclear fractions of bovine thymus, TCF and TNF, contained lower amounts of spermidine compared to SRS (97.14 mg/kg and 143.86 mg/kg, respectively), while TNF had the highest levels of putrescine among all the tested supplements (Fig 2). Desiccated thymus material, on the other hand, was not rich in either putrescine, spermidine or spermine. Neither LNF nor HNF contained appreciable amounts of the most abundant PAs – spermidine and spermine. These data show that a supplement containing wheat germ, which is rich in spermidine,

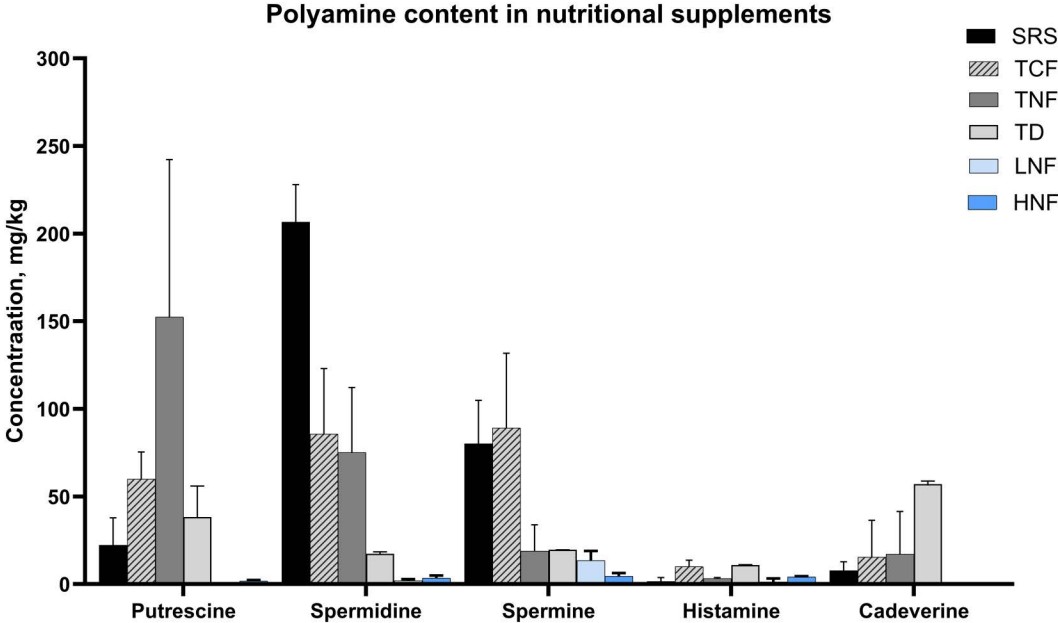

**Fig 2. Polyamine content in nutritional supplements.** Putrescine, spermine, spermidine, histamine and cadaverine were measured in SRS, TCF, TNF, TD, LNF and HNF supplements using HPLC [71]. Data are shown as Mean ± S.E.M.; two distinct lots of materials were analyzed.

as well as cytosolic and nuclear fractions of bovine thymus, can potentially be good sources of spermidine, spermine and putrescine.

## Liposomal particles are present in spermidine-rich SRS, TCF and TNF supplements, but not in freeze dried thymus material

A study by Acosta-Andrade *et al.* (2017) has shown that PAs can induce aggregation of lipid vesicles, such as liposomes [76]. Liposomal delivery of nutrient compounds, as well as drugs, is considered more efficient compared to the delivery of unencapsulated materials because liposomes are stable, lipids-based nanoparticles that are resistant to breakdown through digestion or oxidation, and hence can improve the bioavailability of their cargo [77]. To test whether nutritional supplements containing PAs also contain liposomes we performed Cryo-TEM analysis of the SRS, TCF, TNF and the des-iccated thymus material, TD. Supplements that did not contain appreciable levels of PAs were excluded from further study. We detected liposomes in solutions of SRS, TCF and TNF supplements (Fig 3A–C), while TD was devoid of liposomal structures (Fig 3D).

DLS analysis showed that the average size of liposomes present in TNF was larger compared to SRS and TCF supple-ments (Table 1; S1 Fig A–C). NTA analysis further revealed that SRS contains the highest numbers of liposomal particles compared to TCF and TNF supplements (Table 2). Together, these data demonstrate that the tested supplements com-posed of plant- and bovine tissue-derived materials contain liposomal particles, which are known to protect the encapsu-lated molecules from premature breakdown, thus increasing their bioavailability.

## Liposomal spermidine is highly enriched in TNF supplement compared to TCF and SRS

The presence of liposomal particles in the SRS, TCF and TNF nutritional supplements led us to investigate whether PAs in these supplements were encapsulated within the liposomes. Each of the supplements was examined in 3 distinct ways (see *Materials and Methods*): 1) putrescine, spermidine and spermine were measured in perchloric acid solutions

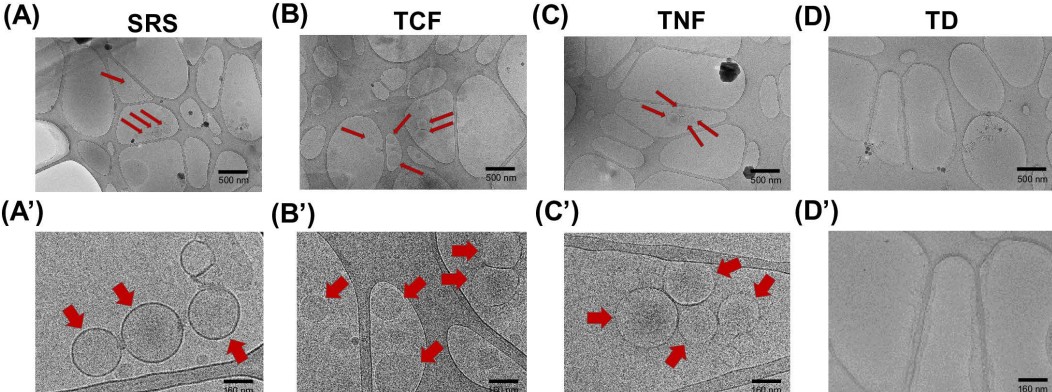

**Fig 3. Liposomes are detected in the SRS, TCF and TNF supplements using Cryo-TEM. (A-D)** Representative images of SRS, TCF, TNF and TD supplements, show the presence of liposomal particles in the SRS, TCF and TNF (A-C; red arrows), but not in TD **(D)**. (A'-D') Higher magnification images shown in **(A-D)**; red arrowheads point to select liposomes. Scale bars in **(A-D)**: 500 nm.

**Table 1. Summary of liposomal particle sizes detected in nutritional supplements (DLS analysis).**

| Supplement | Z-average size | PDI* |
|---|---|---|
| SRS | 206.6 nm | 0.40 |
| TCF | 151.4 nm | 0.24 |
| TNF | 285.2 nm | 0.58 |

*PDI – polydispersity index

**Table 2. Liposomal particle concentrations in individual supplements (NTA analysis).**

| Sample Name | D10* | D50* | D90* | Particle concentration |
|---|---|---|---|---|
| SRS | 113.9 | 174.4 | 307.7 | $5.15 \times 10^8$ particles/mL |
| TCF | 144.3 | 202.4 | 373.4 | $3.55 \times 10^8$ particles/mL |
| TNF | 109.4 | 175.7 | 337.3 | $2.53 \times 10^8$ particles/mL |

*D10, D50 and D90 indicate the points in size distribution where 10%, 50% and 90% of the sample are contained, respectively.[78]

of supplement powders using HPLC (assessing total free PA content), 2) total putrescine, spermidine and spermine were measured in solutions following the destruction of the liposomes (assessing total PA content), and 3) putrescine, spermidine and spermine were measured in destructed liposomes specifically (assessing encapsulated PA content). We found that among the tested supplements, TNF contained the highest total amounts of PAs, measured following the destruction of the liposomes (Total PA content; Table 3). The encapsulation efficiency of PAs was difficult to assess, most likely due to the known vulnerability of liposomes to destruction during the ultracentrifugation process, resulting in losses of certain types of entrapped molecular cargo [79,80]. Relatively low abundance of liposomes in the solutions of each supplement (Table 2; Fig 3) and the limitations of the HPLC assay used (detection limit of 5 µg/mL) could have also contributed to low recovery of the encapsulated PAs.

## Liposomes in the SRS and in thymus gland extracts contain RNA

Partly owing to their positive charge, most PAs within the cells are bound to RNA and exist in PA-RNA complexes [24,38]. We therefore examined whether liposomal particles found in the SRS, TCF and TNF supplements contain RNA. We found

**Table 3. Liposomal PA content in tested nutritional supplements.**

| HPLC assay | Putrescine; ppm* | Spermidine; ppm* | Spermine; ppm* |
|---|---|---|---|
| **Spermidine-Rich Supplement (SRS)** | | | |
| Free PA content | 13.3 | 122.8 | 45.1 |
| Total PA content | 48 | 285 | 95 |
| Liposomal PA content | nd | nd | nd |
| PA encapsulation efficiency | nd | nd | nd |
| **Thymus – Cytosolic Fraction (TCF)** | | | |
| Free PA content | 165.1 | 180.0 | 156.8 |
| Total PA content | 980 | 595 | 404 |
| Liposomal PA content | nd | nd | 5 |
| PA encapsulation efficiency | nd | nd | 1.23% |
| **Thymus – Nuclear Fraction (TNF)** | | | |
| Free PA content | 155.1 | 175.5 | 83.3 |
| Total PA content | 1600 | 1685 | 809 |
| Liposomal PA content | nd | nd | 5 |
| PA encapsulation efficiency | nd | nd | 0.62% |

*Note: 1 ppm = 1 µg/ml; detection limit: 5 ppm; nd – not detected.

**Table 4. Total and liposomal RNA content in tested nutritional supplements.**

| Supplement | Total RNA conc. (ng/mL) | Loaded RNA conc. (ng/mL) | Encapsulation efficiency* |
|---|---|---|---|
| SRS | 49.7143454 | 22.46643454 | 45.19% |
| TCF | 29.53607242 | 21.78467967 | 73.76% |
| TNF | 35.11128134 | 32.15027855 | 91.57% |

*Encapsulation efficiency = Loaded RNA conc./Total RNA conc. X 100.

that RNA was present in all the tested supplements and that RNA in the TNF supplement was predominantly contained within the liposomal particles, evidenced by the highest encapsulation efficiency detected for TNF (91%) (Table 4). Negligible amount of DNA was detected in TCF, and none in the SRS and TNF supplements (S4 Table). These results show that RNA molecules can be potentially delivered by the SRS, TCF and TNF supplements, with most of the RNA contained in the TNF supplement being encapsulated within the liposomal particles.

## Rapid changes in blood PA profiles are induced by oral supplementation of rats with SRS, TCF and TNF

Rapid cyclic interconversion and catabolism of PAs are intricately regulated processes that partly rely on the tissue abundance of each of the PAs and ultimately determine their levels in the blood [63]. To begin investigating whether PAs present in the SRS, TCF and TNF supplements are bioavailable and can impact circulating PA levels, we measured blood PA concentrations at baseline and 5 timepoints within a 6-hour period following supplement administration to rats by oral gavage. Each supplement was administered at a dose that delivered approximately 0.12–0.2 mg/kg BW of spermidine and 0.05–0.13 mg/kg BW of spermine. As expected, we observed rapid changes in the concentrations of PAs in response to the oral delivery of each of the three tested supplements, with these changes being highly similar between the groups (Fig4). First, blood levels of putrescine were moderately decreased 1 hour after the gavage procedure, followed by a gradual increase in blood putrescine levels with each of the three supplements (Fig 4A, B; p < 0.0001); a significant effect

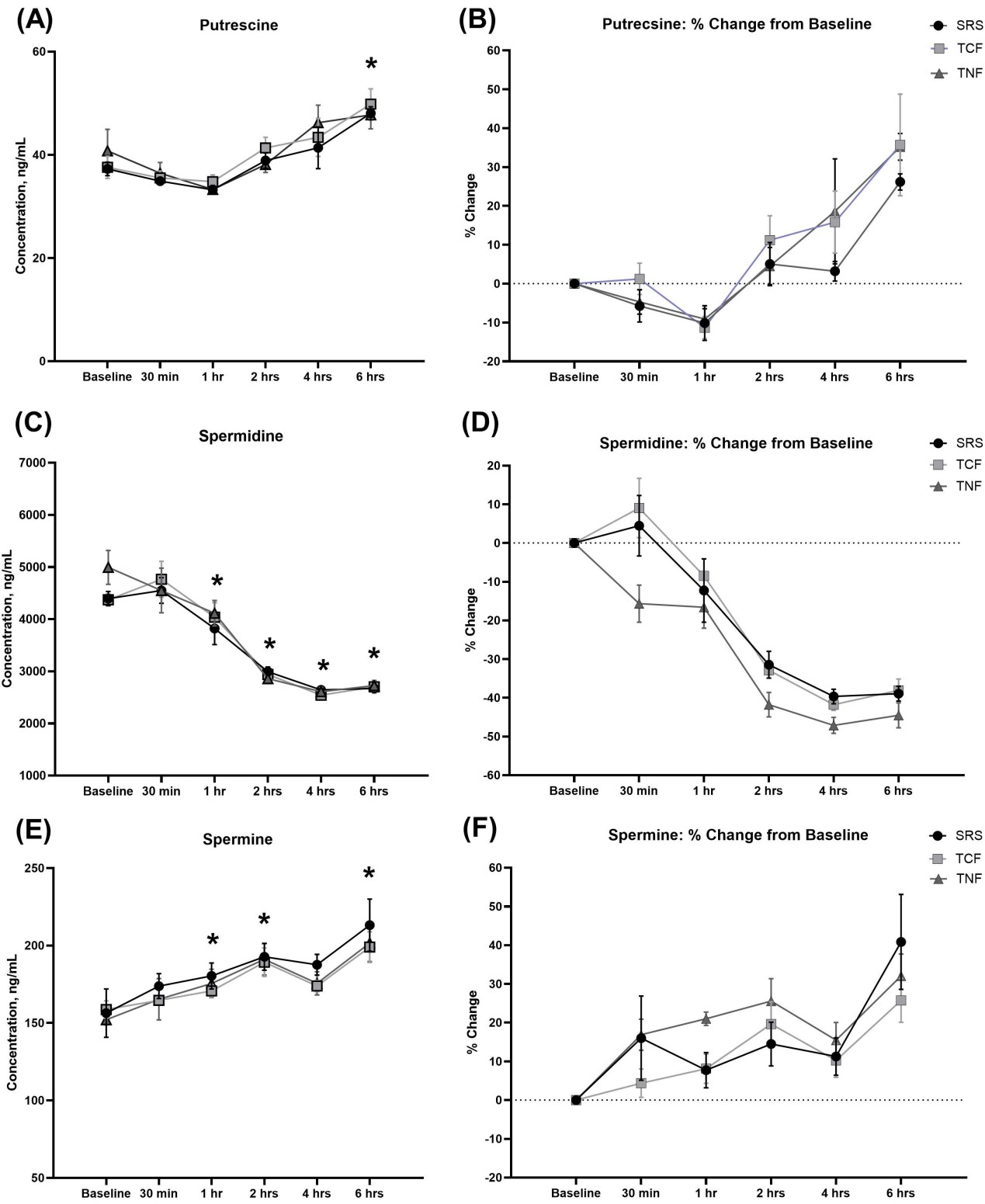

**Fig 4. Acute changes in blood PA concentrations in response to one oral dose of each supplement.** Rats received one oral dose of either SRS, TCF or TNF (see *Materials and Methods*) and blood concentrations of PAs were measured using UPLC-MS/MS at following time intervals: Baseline, 30 min, 1 hr, 2 hrs, 4 hrs and 6 hrs (see *Materials and Methods*). N = 6 animals per group. PA concentrations are shown in (A, C, E) and concentration

changes compared to baseline in (B, D, F). Data are shown as Mean±S.E.M.; * - significant effect of Time variable by Repeated Measures ANOVA and Bonferroni/Dunn post-hoc test, p<0.05.

of time variable was observed at 6 hours after supplement administration compared to baseline (Baseline vs 6 hours: p<0.0001). On the contrary, blood levels of spermidine gradually and significantly decreased during the 6 hours after the oral gavage procedure (Fig 4C, D; p<0.0001), with significant effect of time variable observed at 1–6 hours following supplement administration (Baseline vs 1 hour: p=0.0006; Baseline vs 2 hours: p<0.0001; Baseline vs 4 hours: p<0.0001; Baseline vs 6 hours: p<0.0001). Blood levels of spermine, on the other hand, steadily increased following supplement administration (Fig 4E, F; p<0.0001), with significant effect of time variable observed at 2–6 hours after supplement administration (Baseline vs 2 hours: p<0.0001; Baseline vs 4 hours: p=0.021; Baseline vs 6 hours: p<0.0001). These results show that oral administration of either SRS, TCF or TNF leads to rapid increases in blood concentrations of putrescine and spermine, and a decrease in spermidine.

### Blood and tissue levels of spermidine increase following 28 days of supplementation with TNF and SRS

Having established that the SRS and bovine thymus gland-derived cytosolic and nuclear extracts contain liposomal particles and can deliver liposomal PAs, we hypothesized that prolonged supplementation with the SRS, TCF or TNF would have a distinct effect on PA homeostasis in the blood and in tissues compared to the observed rapid changes in blood PA levels shortly after supplement administration. Sustained delivery of liposomal PAs could potentially affect the enzymes and antizymes involved in PA metabolism and catabolism at the transcriptional and translational levels. We therefore assessed blood and tissue PA concentrations in rats supplemented with the SRS, TCF and TNF for 28 days. Each supplement was administered orally at the same dose of 175 mg/kg BW twice per day. We found that the blood levels of putrescine were reduced by over 15% following sustained supplementation with SRS and TCF, and by over 30% with TNF administration compared to baseline levels (Fig 5A, B; Tables 5–6). On the contrary, blood levels of spermidine increased following sustained SRS and TNF supplementation, with TNF administration resulting in the largest blood spermidine increase of over 22% (Fig 5C, D; Tables 5–6). Finally, blood levels of spermine were modestly reduced only with SRS supplementation (Fig 5E, F; Tables 5–6), tended to increase with TNF treatment, and remained largely unchanged in TCF-supplemented animals compared to baseline.

To determine whether the observed changes in blood PA levels were also accompanied by the differences in tissue PA concentrations between the three supplementation groups, we measured the concentrations of putrescine, spermidine and spermine in brain, heart and liver tissues at the end of a 28-day supplementation period. We found that the levels of all three PAs in brain tissues were not significantly different between the SRS-, TCF- and TNF-supplemented animals (Fig 6A–C; Table 7). However, the levels of spermidine and spermine in the heart and liver tissues differed between the supplementation groups. Specifically, TNF-supplemented animals displayed increased levels of spermidine in the heart compared to TCF-supplemented group (Fig 6E; effect of Treatment variable: p=0.0461 by Bonferroni/Dunn post-hoc test; Table 7). In the liver, spermidine levels were significantly higher in TNF-supplemented animals compared to SRS-supplemented animals (Fig 6H; effect of Treatment variable: p=0.0043 by Bonferroni-Dunn post-hoc analysis; Table 7), accompanied by the increased levels of spermine (Fig 6I; effect of Treatment variable: p=0.0232 by Bonferroni/Dunn post-hoc test; Table 7). Liver putrescine levels remained unchanged between the groups (Fig 6G; Table 7). Importantly, no significant changes in spermine/spermidine ratios, which are known to decrease in diseased states, were observed in the blood or tissues of TNF-supplemented animals, while blood spermine/spermidine ratio decreased with SRS supplementation (S2 Fig) [18,81]. Thus, among the three tested supplements, TNF is most potent at raising the blood and tissue levels of spermidine and spermine without affecting the spermine/spermidine ratios.

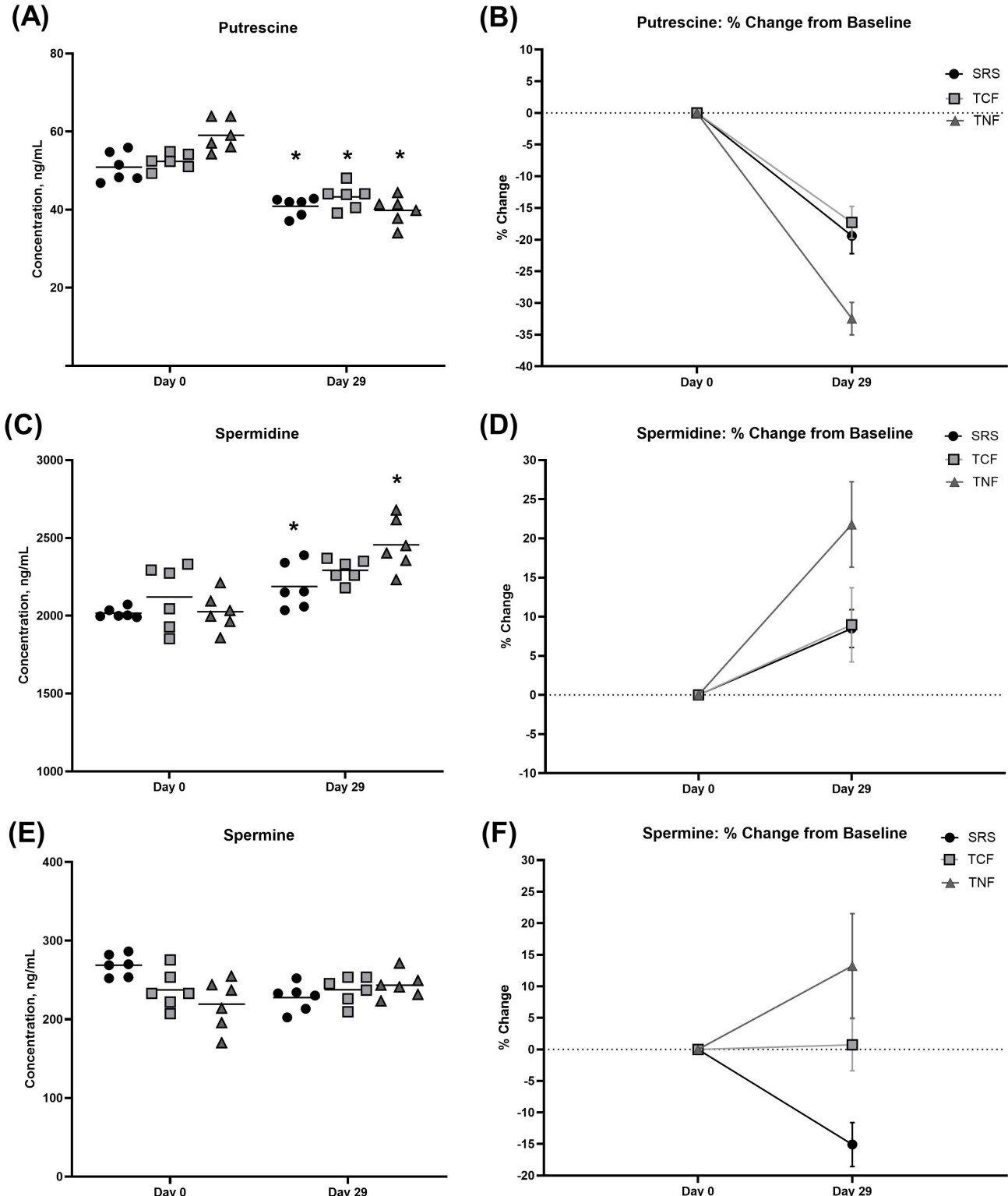

**Fig 5. Changes in blood PA concentrations following 28 days of supplementation with SRS, TCF and TNF.** (A, C, **E**) Blood concentrations of putrescine were significantly reduced in SRS-, TCF- and TNF-supplemented animals **(A)**, spermidine levels significantly increased following

supplementation with SRS and TNF (C) and spermine levels did not change significantly between the groups (E) on Day 29 compared to baseline (Day 0). Data are shown as minimum/maximum box plots, N = 6 animals per group. (B, D, **F**) Changes in blood putrescine **(B)**, spermidine (D) and spermine (F) from baseline are shown as individual values and the mean; * - p < 0.05 by Paired Student's t-test (please, see Table 5).

**Table 5. Blood PA concentrations in individual animals at baseline and following 28 days of supplementation.**

| Animal # | SRS | | | TCF | | | TNF | | |
|---|---|---|---|---|---|---|---|---|---|
| | Baseline, ng/mL | | | | | | | | |
| | PUT | SPD | SPM | PUT | SPD | SPM | PUT | SPD | SPM |
| 1 | 55.91 | 2002.76 | 270.00 | 54.19 | 1928.07 | 233.04 | 59.06 | 1963.27 | 179.21 |
| 2 | 54.76 | 1990.87 | 268.90 | 54.86 | 2331.53 | 207.31 | 54.28 | 2094.59 | 254.94 |
| 3 | 48.08 | 2035.59 | 253.57 | 49.32 | 2274.44 | 233.04 | 63.92 | 2033.69 | 237.15 |
| 4 | 46.84 | 1999.43 | 282.32 | 51.04 | 2293.47 | 253.57 | 57.05 | 2212.58 | 243.99 |
| 5 | 51.52 | 2072.70 | 286.43 | 52.47 | 1852.89 | 222.09 | 63.92 | 1996.58 | 195.81 |
| 6 | 48.27 | 1996.58 | 252.20 | 52.38 | 2044.63 | 275.48 | 56.10 | 1858.60 | 214.15 |
| **Min** | **46.84** | **1996.58** | **252.20** | **51.04** | **1852.89** | **207.31** | **54.28** | **1858.60** | **179.21** |
| **Max** | **55.91** | **2072.70** | **286.43** | **54.86** | **2293.47** | **275.48** | **63.92** | **2212.58** | **254.94** |
| Day 29, ng/mL | | | | | | | | | |
| Animal # | PUT | SPD | SPM | PUT | SPD | SPM | PUT | SPD | SPM |
| 1 | 55.91 | 2035.593 | 202.52 | 48.08 | 2179.28 | 209.633 | 44.36 | 2678.85 | 249.47 |
| 2 | 54.76 | 2155.491 | 252.20 | 44.07 | 2260.16 | 253.573 | 34.05 | 2355.32 | 243.44 |
| 3 | 48.08 | 2388.624 | 233.04 | 44.07 | 2260.16 | 253.573 | 41.40 | 2617.00 | 231.67 |
| 4 | 46.84 | 2058.907 | 213.47 | 43.88 | 2331.53 | 245.36 | 41.40 | 2231.62 | 241.25 |
| 5 | 51.51 | 2341.046 | 234.41 | 40.54 | 2350.56 | 226.196 | 37.77 | 2402.90 | 223.46 |
| 6 | 48.27 | 2150.733 | 230.30 | 39.11 | 2369.59 | 202.515 | 39.80 | 2450.48 | 271.37 |
| **Min** | **48.27** | **2058.907** | **202.52** | **39.11** | **2179.28** | **202.515** | **34.05** | **2231.62** | **223.46** |
| **Max** | **55.91** | **2388.624** | **252.20** | **48.08** | **2369.59** | **253.573** | **44.36** | **2678.85** | **271.37** |

PUT, putrescine; SPD, spermidine; SPM, spermine.

**Table 6. Blood PA concentrations at baseline and following 28 days of supplementation.**

| Supplement | Polyamine; ng/mL | n | D0 (Baseline) | D29 | Paired Student's t-test P-value |
|---|---|---|---|---|---|
| | | | Mean ± SD | Mean ± SD | |
| SRS | Putrescine | 6 | 50.896 ± 3.787 | 40.859 ± 2.360 | 0.0020* ↓ |
| | Spermidine | 6 | 2016.324 ± 31.802 | 2188.399 ± 145.610 | 0.0182* ↑ |
| | Spermine | 6 | 268.904 ± 14.157 | 227.656 ± 17.422 | 0.0090* ↓ |
| TCF | Putrescine | 6 | 52.375 ± 2.028 | 43.292 ± 3.148 | 0.0011* ↓ |
| | Spermidine | 6 | 2120.838 ± 206.178 | 2291.882 ± 71.757 | 0.1185 |
| | Spermine | 6 | 237.421 ± 24.039 | 237.580 ± 17.221 | 0.9886 |
| TNF | Putrescine | 6 | 59.056 ± 4.073 | 39.796 ± 3.551 | 0.0001* ↓ |
| | Spermidine | 6 | 2026.553 ± 120.331 | 2456.027 ± 166.685 | 0.0094* ↑ |
| | Spermine | 6 | 219.375 ± 32.208 | 243.444 ± 16.485 | 0.1763 |

* - statistically significant change, p < 0.05; ↑ - upward change; ↓ - downward change

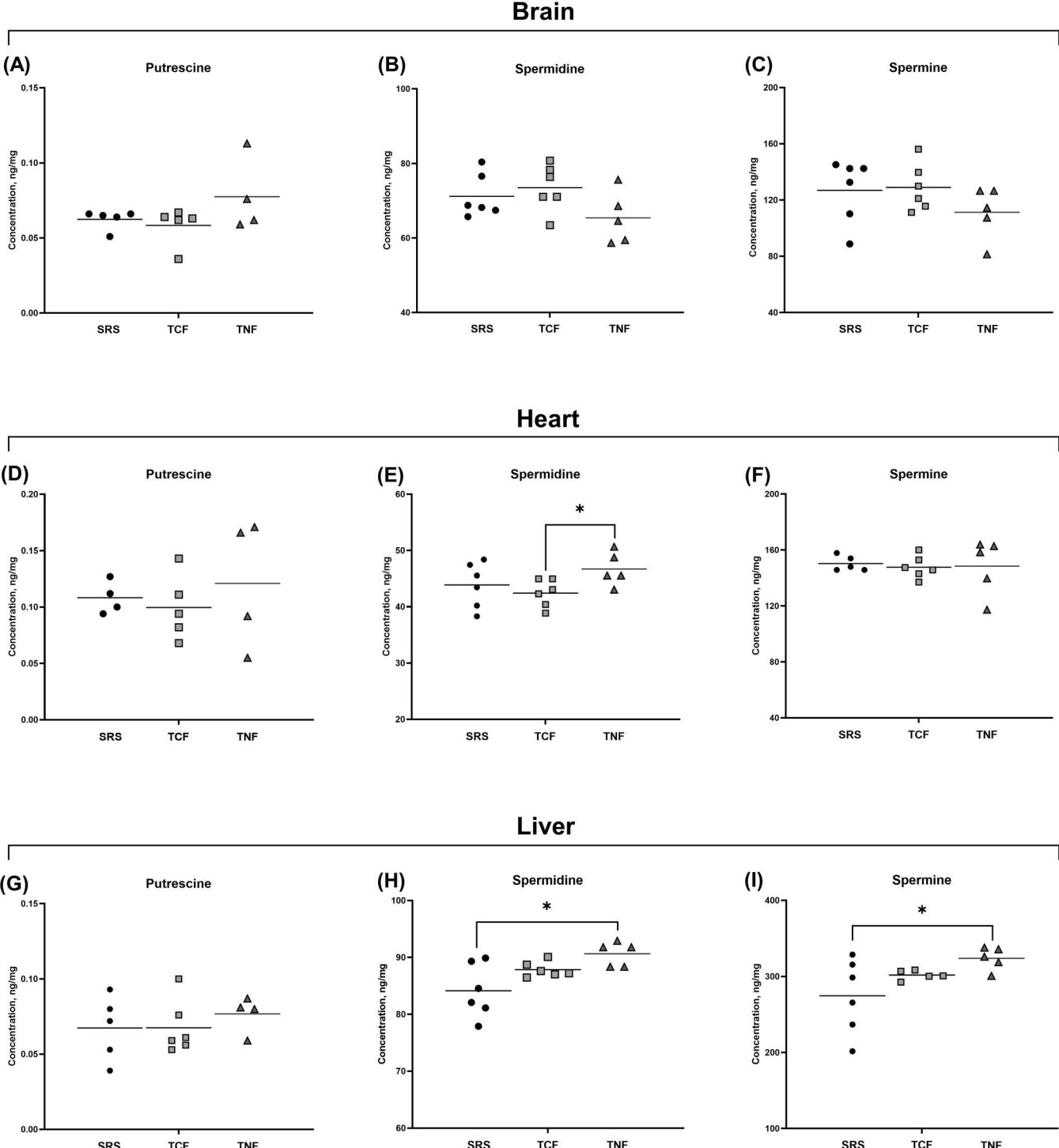

**Fig 6. Comparison of PA concentrations in the brain, heart and liver tissues of rats supplemented with SRS, TCF and TNF for 28 days.**
Concentrations of putrescine, spermidine and spermine were measured in tissues of supplemented animals at the end of the 28-day supplementation period. **(A-C)** In brain tissues, no differences in PA levels were observed between the supplementation groups. **(D-F)** In heart tissues, TNF supplementation resulted in increased spermidine levels compared to TCF-supplemented animals. (G-I) In the liver, spermidine and spermine levels were higher in TNF-supplemented compared to SRS-supplemented animals. Data are shown as individual values and the mean; N = 4-6 animals per group. * - P < 0.05 by Bonferroni/Dunn post-hoc analysis (see Table 6).

**Table 7. PA concentrations in the brain, heart and liver tissues following 28 days of supplementation.**

| Supplement | Polyamine | n | Brain; D29 Mean±SD | n | Heart; D29 Mean±SD | n | Liver; D29 Mean±SD |
|---|---|---|---|---|---|---|---|
| SRS | Putrescine, ng/mL | 5 | 0.062±0.006 | 4 | 0.108±0.015 | 5 | 0.067±0.022 |
| | Spermidine, ng/mL | 6 | 71.172±5.868 | 6 | 43.893±4.013 | 6 | 84.145±4.738 |
| | Spermine, ng/mL | 6 | 126.981±22.731 | 5 | 150.270±5.398 | 6 | 274.452±49.015 |
| TCF | Putrescine, ng/mL | 5 | 0.058±0.013 | 5 | 0.100±0.029 | 6 | 0.068±0.018 |
| | Spermidine, ng/mL | 6 | 73.487±6.278 | 6 | 42.434±2.451 | 6 | 87.856±1.328 |
| | Spermine, ng/mL | 6 | 128.989±16.786 | 6 | 147.696±7.978 | 5 | 301.719±6.283 |
| TNF | Putrescine, ng/mL | 4 | 0.078±0.025 | 4 | 0.121±0.057 | 4 | 0.077±0.012 |
| | Spermidine, ng/mL | 5 | 65.373±6.997 | 5 | 46.723±3.001[b] ↑ | 5 | 90.647±2.136[a] ↑ |
| | Spermine, ng/mL | 5 | 111.340±18.615 | 5 | 148.408±19.910 | 5 | 323.840±15.047[a] ↑ |

[a] – significant difference from SRS, p<0.05 by Bonferroni/Dunn post-hoc analysis; [b] – significant difference from TCF, p<0.05 by Bonferroni/Dunn post-hoc analysis; ↑- upward change.

## Discussion

Diets rich in PAs are considered beneficial for supporting optimal health and may promote longevity [11,12,17,82]. Considering that endogenous PA synthesis may gradually diminish with age, along with the increasing PA catabolism, reduction in PA production by the gut bacteria and compromised intestinal absorption during aging, nutritional supplementation with PA-containing products and increased dietary intake of PA-rich foods are plausible approaches towards ensuring adequate availability of PAs in older adults [1,9,14,16,82,83]. However, elevating the circulating levels of PAs in the blood, which are thought to reflect their tissue levels as well, through dietary supplementation appears to be challenging [5,53,58,60,82]. In this study, we describe a novel finding that the cytoplasmic and nuclear extracts of bovine thymus gland, TCF and TNF, as well as a nutritional supplement containing wheat germ, SRS, can all serve as the sources of dietary PAs, with TNF being most potent at elevating blood and tissue levels of spermidine.

Among the materials tested in this study, TNF elicited the largest increase in blood spermidine levels (22%) after 28 days of supplementation. This effect of TNF is not likely to be mediated strictly by the total amounts of PAs it contains, but to their liposomal encapsulation in this particular type of thymus gland extract. Indeed, we found that the concentration of total spermidine is the highest in the SRS material, which contains wheat germ (approximately 207 mg/kg) – nearly double the amounts detected in TCF and TNF (approximately 86 mg/kg and 75 mg/kg, respectively). This finding is consistent with the reports that wheat germ is a good dietary source of PAs [47]. Between the bovine thymus gland-derived supplements, however, TNF contains the lowest amount of spermine, but the highest amount of PA precursor – putrescine (approximately 152 mg/kg). Surprisingly, the nuclear fraction extracts of two other bovine organs we tested, liver and heart, do not contain appreciable amounts of putrescine, spermidine or spermine, suggesting that PA content in these extracts may be low due to distinct cellular compositions of liver and heart tissues compared to thymus, or due to losses during the nuclear material extraction procedure. Nevertheless, this study is the first to report on the presence of PAs in nutritional supplements composed of natural ingredients, including bovine glandular tissue extract and wheat germ, and the effects of these supplements on PA homeostasis *in vivo*.

Nutritional supplementation with PAs is typically approached using plant extracts, including bacterial fermentation products (i.e., wheat germ extract; natto), or synthetic PA forms [1,53,58,59]. Liposomal forms of spermidine are also available as supplements and can potentially increase its bioavailability, but to our knowledge, their effectiveness in supporting blood or tissue PA levels has yet to be reported. In this study, we show that SRS, TCF and TNF contain lipid nanoparticles that fit the characteristics of liposomes – bilayer vesicles composed of lipids and phospholipids, which can

spontaneously form in aqueous solutions and primarily encapsulate hydrophilic molecules present in such solutions (i.e., positively charged polyamines) [84]. Interestingly, the concentration of lipid nanoparticles in the SRS ($5.15 \times 10^8$ particles/mL) is nearly double their concentration in the TNF supplement ($2.53 \times 10^8$ particles/mL), while nanoparticle average size is the largest in TNF among the tested supplements – 285 nm compared to 151 nm in TCF and 207 nm in the SRS. The size distribution of the observed nanoparticles (151–258 nm in diameter), combined with their presence in the extracts of the intracellular cytoplasmic and nuclear materials, excludes the possibility of them being either exosomes, which only reach approximately 100 nm is diameter, or other types of extracellular, secreted microvesicles [85,86]. The PDI values, which serve as measures of liposome quality during their production, were below 0.7 for all three tested supplements, indicating that liposomal particles contained in these materials are relatively uniform with respect to their sizes [87]. Unlike the artificial liposomes designed for the improvement of drug or vitamin/nutrient formulations and delivery, which are typically produced using defined ratios of phospholipids isolated from soy, eggs, milk, bacteria or synthetic phospholipids [84,88–91], self-assembly of liposomes found in the SRS, TCF and TNF is likely to be induced during the extraction process from the lipids that naturally exist within the plant and animal tissue intracellular materials. Hence, we observed that the desiccated (freeze dried) thymus material, which was not subjected to the extraction process, does not contain liposomal particles. While the sizes of liposomes observed in the SRS, TCF and TNF supplements (> 151 nm) suggest that they are likely to enter cells via endocytosis [87], and micropinocytosis specifically, further studies are needed to determine the precise lipid compositions of these particles, their content and physiological properties.

Because PAs can promote aggregation and fusion of liposomes, and often exist within the molecular complexes containing RNA, it is not surprising that these lipid particles are found in PA-rich extracts of plant and thymus gland materials used in this study, and also contain RNA in addition to PAs [24,76,92]. Yet, the amounts of PAs and RNA encapsulated in the liposomes vary greatly between the supplements we examined. We show that the concentrations of liposomal PAs, and spermidine in particular, are severalfold higher in the TNF supplement compared to the SRS and TCF materials, and that over 90% of the RNA detected in TNF material is encapsulated within the liposomes. Therefore, it is likely that the delivery of larger amounts of liposomal, as opposed to free PAs, underlies the capacity of TNF supplement to elevate blood and tissue levels of spermidine following a 28-day supplementation period compared to the SRS and TCF. The presence of RNAs within the tested supplements, on the other hand, confirms that lipid nanoparticle-RNA formulations are quite stable [93–95], and that a range of bioactive molecules, including PA-RNA complexes, are encapsulated in the liposomes derived from the animal and plant material extracts. Contrary to appreciable levels of RNA, we detected a small amount of DNA in the TCF, and none in the SRS and TNF supplements, possibly due to DNA degradation and losses in aqueous solutions during plant and animal material extraction processes, especially if salts were present (i.e., salt-dependent proprietary extraction method of nuclear material) [96,97]. We confirmed that larger amounts of RNA compared to DNA are found across multiple nuclear fraction extracts of bovine tissues prepared using the same methodology as TNF. Thus, the technology utilized in the extraction of plant and animal tissue materials, and the types of lipids and soluble molecules present in various types of biological matrices, can determine the probability of liposomal particle assembly and the molecular cargo of the liposomes produced, including the final molecular composition of the product [98]. This study highlights the unique properties of nutritional supplements composed of plant and animal tissue-derived materials and their capacity to deliver a range of bioactive molecules within naturally occurring matrices, including liposomal particles.

The bioavailability of PAs contained in the SRS, TCF and TNF is first evidenced by our finding that one dose of each supplement exerts pronounced and immediate effects on blood PA levels. Within 6 hours following supplement administration at doses that in humans would be equivalent to 1.29–2.12 mg of spermidine, we observed gradual and significant increases in blood putrescine and spermine levels (approximately 30%), while spermidine levels gradually declined by approximately 40%. Lack of significant differences in the directions and kinetics of blood PA concentration changes among the three supplementation groups could be explained by comparable concentrations of free, non-liposomal spermine and spermidine in tested supplements that could drive the rapid changes in blood PA levels. The observed increase in

blood spermine levels in SRS-, TCF- and TNF-supplemented animals is consistent with previous studies that reported its changes in the blood in response to PA-rich extracts or foods [58,82,99,100]. In addition, preferential association of spermine with cellular membranes or nucleic acids *in vivo* may contribute to the lower affinity of SSAT enzyme, which regulates cellular PA content, for spermine compared to its acetylated form, N1-acetylspermine, possibly leading to the temporary increase in spermine we observed in this part of the study [101]. An increase in blood putrescine levels, on the other hand, has been demonstrated to occur temporarily in response to spermidine administration in rats, peaking at 6 hours, and to be mediated by PA-dependent induction of SSAT and subsequent conversion of N1-acetylspermidine to putrescine by polyamine oxidase (PAO) [102]. This mechanism could partly explain the temporary depletion of spermidine and concomitant increase in putrescine during the 6-hour period following supplement administration.

The increases in blood spermidine levels after 28 days of supplementation with TNF and SRS at doses delivering relatively low total amounts of spermidine, equivalent to approximately 0.53 mg and 0.76 mg per day in humans, respectively, further support the bioavailability of PAs present in these materials. This finding adds to the current understanding of dietary PA metabolism, which is based on studies that reported increases in blood or tissue spermidine levels only with substantially higher spermidine doses or longer supplementation periods [5,53,58,59,100,102]. The observed reduction in blood putrescine levels across the supplementation groups could be explained by possible inhibition of ornithine decarboxylase (ODC) enzyme by exogenous PAs through Antizyme 1-mediated degradation, leading to attenuated putrescine synthesis [103]. S-adenosylmethionine decarboxylase (AdoMetDC), an enzyme catalyzing the production of decarboxylated S-adenosylmethionine (dcAdoMet), which serves a rate limiting role in the synthesis of spermidine and spermine [104,105], and ODC can both be inhibited by spermine and spermidine as part of the mechanism regulating intracellular PA homeostasis, leading to reduced endogenous synthesis of spermine/spermidine and a reduction in putrescine, respectively [106]. SSAT, a key enzyme responsible for the conversion of spermine to spermidine, and spermidine to putrescine, is activated/induced by spermine and spermidine themselves, and may prevent the rise in blood levels of spermine [101,102]. At the same time, the activities of PAO, which can use spermine, N1-acetylspermine and N1-acetylspermidine as substrates, but not spermidine, and spermine oxidase (SMO), can potentially contribute to the elevation in blood levels of spermidine, and not spermine, we observed following 28 days of supplementation with TNF [102,107,108]..

The concentrations of PAs in the blood and tissues are linked, such that the variation in health status and diets across individuals contributes to varying blood PA levels observed in human and animal studies, with and without dietary supplementation regimens [5,82]. For example, inflammatory conditions and cancer, among other pathologies, can both lead to dysregulation of PA homeostasis, reflected by either diminished or elevated blood and tissue levels of individual PAs [18,82,109]. Variability in dietary intakes of PA-containing foods can further contribute to inter-personal differences in circulating PA levels [14]. Consistent with this, we detected variability in baseline blood PA levels among the animal groups undergoing 28-day supplementation regimens. Nevertheless, we demonstrate that TNF supplement is superior to the SRS and TCF in increasing blood spermidine concentration compared to baseline values, and in elevating the levels of spermidine in the heart and liver tissues. Consistent with the reports of poor brain penetrance of circulating PAs, we did not detect differences in brain PA levels among the supplementation groups [110,111]. Similarly, we did not detect significant changes in spermine/spermidine ratios, which can reflect diseased states [18,19,81], in the blood and tissues of TNF-supplemented animals – TNF elicited a moderate increase in blood spermine levels concurrently with spermidine, and a significant increase in liver spermine levels compared to the SRS group. On the contrary, an increase in blood spermidine levels with SRS supplementation was not accompanied by an increase in blood spermine levels, leading to a reduction in blood spermine/spermidine ratio in this group.

Even though it remains to be determined whether liposomal particles present in each of the tested supplements are transported intact across the intestinal enterocytes or taken up within the Pyer's patches, as opposed to releasing their content in the intestine [112], TNF, and to a lesser extent SRS, are potent at delivering bioavailable forms of PAs that, even at relatively low doses, increase blood spermidine levels in 28 days. Considering that elevating the circulating levels

of PAs through supplementation has been a challenge, oral delivery of these bioactive molecules within the matrices inherent to animal organs that are naturally rich in PAs, such as the extract of the thymus gland described in this study, appears to be an advantage.

Adequate dietary supply of PAs is important for ameliorating inflammatory conditions, including systemic chronic inflammation prevalent among aging individuals and leading to decline in health and age-related diseases [28,113–115]. From improving gut barrier integrity and obesity indices, to direct inhibition of inflammatory diseases and proinflammatory status, dietary PA supplementation was shown to be effective in animal models and in humans [60,115,116]. Likewise, we previously demonstrated that in the context of CNS injury, the neuroprotective role of TNF supplement is accompanied by substantial changes in the expression of genes implicated in the regulation of inflammation, autophagy and tissue repair – the processes that are highly relevant to the maintenance of good health during aging [70]. Considering the proposed general decline in PA status among aging individuals, low intake of organ meats in Western diet, and current difficulties in increasing circulating levels of spermidine through dietary means, nutritional supplementation with TNF could be a plausible approach towards delivering bioavailable forms of PAs and supporting health during aging. The effectiveness of TNF in modulating the circulating levels of PAs in humans remains to be evaluated.

## Conclusions

In this study, we demonstrated that a nutritional supplement containing a nuclear fraction extract of bovine thymus gland (thymus nuclear fraction, TNF) is effective in elevating blood and tissue levels of spermidine within a 28-day treatment period. Since PA-rich diet plays a vital role in preventing age-related diseases and promoting longevity, maintenance of the adequate dietary supply of PAs during aging is of great importance. To date, dietary supplementation with PA-rich products and foods, even if prolonged, has proven to be ineffective in increasing blood and tissue levels of spermine and spermidine. This study demonstrates that spermidine contained in TNF supplement is encapsulated in naturally occurring liposomal particles, which may underly the effectiveness of TNF in increasing spermidine levels in the blood and in tissues. Moreover, we show that acute versus chronic oral administration of polyamines exert distinct effects on the metabolism of putrescine, spermidine and spermine in the blood.

## Supporting information

**S1 Table. Protocol for measuring encapsulated PAs in nutritional supplements using HPLC.**
(DOCX)

**S2 Table. LC elution gradients.**
(DOCX)

**S3 Table. MS parameters.**
(DOCX)

**S4 Table. DNA concentrations in nutritional supplements.**
(DOCX)

**S1 Fig. Distribution of liposomal particle sizes in the SRS, TCF and TNF materials, assessed using dynamic light scattering (DLS) analysis.** The dimensions of liposomes and their polydispersity indices (PDI) are shown for the SRS (A), TCF (B) and TNF (C).
(TIF)

**S2 Fig. Spermine/spermidine ratios in the blood and tissues of animals supplemented with SRS, TCF and TNF for 28 days.** (A) Blood spermine/spermidine ratio was significantly reduced in SRS-supplemented animals compared to

baseline, while no differences in spermine/spermidine ratios among the supplementation groups were detected in tissues: brain (B), heart (C) and liver (D). Data are shown as individual values and the mean; N = 4–6 animal per group. * - p < 0.0001 for effect of Time variable by Bonferroni/Dunn post-hoc analysis.
(TIF)

## Acknowledgments

We would like to thank Creative Biostructure for their help in detecting and characterizing the liposomes and their content, and Creative Proteomics for measuring PA concentrations in animal blood and tissues.

## Author contributions

**Conceptualization:** Bassem F. El-Khodor, Ashley Dominique.

**Formal analysis:** Bassem F. El-Khodor, Ashley Dominique, Taleen Hanania, Melville Osborne.

**Investigation:** Bassem F. El-Khodor, Taleen Hanania, Melville Osborne.

**Methodology:** Bassem F. El-Khodor, Ashley Dominique.

**Visualization:** Bassem F. El-Khodor, Natalia Surzenko, Ashley Dominique.

**Writing – original draft:** Bassem F. El-Khodor, Natalia Surzenko, Ashley Dominique.

**Writing – review & editing:** Bassem F. El-Khodor, Ashley Dominique.

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
