## [Decision Letter · Decision Letter 0]

28 May 2025

PONE-D-25-09540Nutritional Supplement Containing a Nuclear Fraction of Bovine Thymus Gland Increases the Circulating Levels of SpermidinePLOS ONE

Dear Dr. El-Khodor,

Thank you for submitting your manuscript to PLOS ONE. After careful consideration, we feel that it has merit but does not fully meet PLOS ONE’s publication criteria as it currently stands. Therefore, we invite you to submit a revised version of the manuscript that addresses the points raised during the review process.

We look forward to receiving your revised manuscript.

Kind regards,

Ajit Prakash, PhD

Academic Editor

PLOS ONE

Journal Requirements:

“This study was funded by Standard Process, Inc.”

“N.S., A.D., and B.F.E-K. are current employees of Standard Process Inc.; all other authors declare no conflicts of interest.”

We note that one or more of the authors are employed by a commercial company: Standard Process Inc.

6. We note that your Data Availability Statement is currently as follows: All relevant data are within the manuscript and in Supporting Information files.

7. We note that you have included the phrase “data not shown” in your manuscript. Unfortunately, this does not meet our data sharing requirements. PLOS does not permit references to inaccessible data. We require that authors provide all relevant data within the paper, Supporting Information files, or in an acceptable, public repository. Please add a citation to support this phrase or upload the data that corresponds with these findings to a stable repository (such as Figshare or Dryad) and provide and URLs, DOIs, or accession numbers that may be used to access these data. Or, if the data are not a core part of the research being presented in your study, we ask that you remove the phrase that refers to these data.

Reviewers' comments:

Reviewer's Responses to Questions

**Comments to the Author**

1. Is the manuscript technically sound, and do the data support the conclusions?

Reviewer #1: Yes

Reviewer #2: Partly

2. Has the statistical analysis been performed appropriately and rigorously? 

Reviewer #1: Yes

Reviewer #2: Yes

3. Have the authors made all data underlying the findings in their manuscript fully available?

Reviewer #1: Yes

Reviewer #2: Yes

4. Is the manuscript presented in an intelligible fashion and written in standard English?

Reviewer #1: Yes

Reviewer #2: Yes

5. Review Comments to the Author

Reviewer #1: Comments

The manuscript titled “Nutritional Supplement Containing a Nuclear Fraction of Bovine Thymus Gland Increases the Circulating Levels of Spermidine” presents a comprehensive study of the effectiveness of nutritional supplement containing a nuclear fraction extract of bovine thymus gland (thymus nuclear fraction, TNF) in elevating blood and tissue levels of spermidine within a 28-day treatment period as compared to SRS and TCF. The authors have extensively reviewed the topic and provided necessary references. The quality of the figures and tables is satisfactory.

The study is robustly supported by Cryogenic Transmission Electron Microscopy, liposomal particle size distribution analysis utilizing dynamic light scattering, Nanoparticle Tracking Analysis, and various other analytical techniques.

However, there are a few queries related to the manuscript that need to be addressed.

1. Briefly mention why it is challenging to elevate circulating PA levels through dietary supplements.

2. Under topic, 3.3 Liposomal spermidine is highly enriched in TNF supplement compared to TCF and SRS, to improve the detection and recovery of encapsulated polyamines, other experimental strategies like ultracentrifugation, HPLC-MS/MS can also be explored. Explain why other strategies were not considered.

3. In this study, did you measure any functional outcomes like inflammation or oxidative stress?

4. The blood PA concentration studies have been performed in 6-hour time frame, how did you ascertain there might not be any elevation in levels post 6 hours?

5. Is there any direct evidence that ODC activity was suppressed?

Reviewer #2: In section 3.2, the authors mentioned the presence of liposomal nanoparticles (NPs) in spermidine-rich SRS, TCF and TNF supplements. However, there is no evidence of toxicity measurement or analysis shown by the authors. The author should also consider detecting other NPs.

The authors should also provide an illustrative model on a potential mechanism of how nuclear fraction of bovine thymus supplement can increase spermidine levels. Also, the authors should mention how the current described strategy will provide a way to decrease age related diseases.

In animal models the rat ages and relevant details need to be mentioned.

6. PLOS authors have the option to publish the peer review history of their article (what does this mean? ). If published, this will include your full peer review and any attached files.

**Do you want your identity to be public for this peer review?** For information about this choice, including consent withdrawal, please see our Privacy Policy .

Reviewer #1: **Yes: ** Harpreet Kaur

Reviewer #2: No

---

## [Author Response · Author response to Decision Letter 1]

22 Jul 2025

June 27, 2025

Re: PONE-D-25-09540

Dear Dr. Prakash,

We thank you and the Reviewers for the detailed feedback on our manuscript, titled “Nutritional Supplement Containing a Nuclear Fraction of Bovine Thymus Gland Increases the Circulating Levels of Spermidine”. We carefully considered the insightful comments we received, and believe that the changes we incorporated into the revised version of the manuscript have significantly improved its quality. Below we list our responses to each Reviewer’s question or comment and the specific revisions we made to the manuscript. The changes we incorporated are highlighted in the manuscript text. We hope you find the revised version of our manuscript suitable for publication.

Reviewer 1 comments:

1. Briefly mention why it is challenging to elevate circulating PA levels through dietary supplements.

Authors’ response:

We thank the Reviewer for the suggestion of adding a more detailed explanation of why polyamine availability from the diet may be low. In the Introduction section of the manuscript, we mention that “PAs that come from either the diet or gut microflora are absorbed intact in the gut lumen, but their concentrations drop from millimolar to low micromolar ranges rapidly after ingestion through the mechanisms that remain poorly understood, making PAs “high supply and low utilization” compounds”, and cite the references (lines 69-72).

We further revised the Introduction section to add information and references on potential mechanisms that may limit the availability of dietary PAs as follows (lines 72-78):

Even though specialized carriers have been proposed to mediate PA transport across intestinal enterocytes, their affinity for PAs may differ based on their location at the apical versus basal membranes of enterocytes, while PAs that are absorbed are quickly distributed across tissues with high proliferative demands, including the cells of the gastrointestinal tract itself. In addition, PAs that reach rat tissues 6 hours following exogenous PA administration mediate a 30-300 fold increase in the activity of spermidine/spermine N-1-acetyltransferase (SSAT), an enzyme involved in the conversion of higher PAs back to putrescine, thereby limiting the availability of PAs.

2. Under topic, 3.3 Liposomal spermidine is highly enriched in TNF supplement compared to TCF and SRS, to improve the detection and recovery of encapsulated polyamines, other experimental strategies like ultracentrifugation, HPLC-MS/MS can also be explored. Explain why other strategies were not considered.

Authors’ response:

We appreciate the Reviewer bringing to our attention a possibility of using additional methodology for the analysis of liposomal forms of polyamines. In the analysis of lyposomal polyamine amounts, we used well-validated HPLC methodology of polyamine detection, performed by Creative Biostructure LLC, and referenced it in the Materials and Methods section.

A different detection methodology was not explored because the accurate and absolute amounts of lyposomal versus total polyamines in each supplement are unlikely to directly translate to their bioavailability. The three nutritional supplements tested in this study represent distinct complex biological matrices (i.e., nuclear versus cytoplasmic extracts of various tissues) containing a variety of bioactive components (i.e., lipids, proteins, RNA etc). We would therefore not expect polyamines, which themselves are known to enhance the absorption of micromolecules, to be bioavailable to the same extent when delivered within distinct matrices.

On the other hand, animal blood and tissue levels of PAs were measured using a UPLC-MS/MS method, performed by Creative Proteomics – a laboratory specializing in detection of polyamines and other metabolites.

3. In this study, did you measure any functional outcomes like inflammation or oxidative stress?

Authors’ response:

The functional outcomes of TNF supplementation were not measured in this particular study. We did, however, demonstrate the benefits of TNF supplementation for the functional recovery from traumatic brain injury in an earlier study, including its effects on cognitive performance and the expression of biomarkers of neuroinflammation and autophagy.1 This study is discussed in our manuscript. In addition, we have addressed the functional outcomes of TNF supplementation, and its effects on longevity, neuronal excitability and protection from oxidative stress specifically, in two separate studies that use the aging mouse and C. elegans models. These studies yielded positive results and are currently in preparation for publication this year. Briefly, because TNF is likely to contain a range of diverse bioactive molecules in addition to polyamines, we believe that its benefits are pleiotropic. However, we believe that the understanding that distinct types of bovine tissue extracts can be the sources of bioavailable polyamines is of value to readers from many disciplines.

Reference:

1. Surzenko N, Bastidas J, Reid RW, Curaba J, Zhang W, Bostan H, Wilson M, Dominique A, Roberson J, Ignacio G, Komarnytsky S, Sanders A, Lambirth K, Brouwer CR, El-Khodor BF. Functional recovery following traumatic brain injury in rats is enhanced by oral supplementation with bovine thymus extract. FASEB J. 2024 Feb 15;38(3):e23460. doi: 10.1096/fj.202301859R. PMID: 38315443.

4. The blood PA concentration studies have been performed in 6-hour time frame, how did you ascertain there might not be any elevation in levels post 6 hours?

Authors’ response:

We thank the Reviewer for the insightful question. Studies of radiolabeled polyamine clearance from blood suggested that their levels drop 67-89% within 10 minutes of injection into rat major artery and vein in vivo, and continue to decline rapidly in the next 1.5 hours. Similarly, blood concentrations of polyamines after one oral dose were shown to peak at 1.5 hour post administration and rapidly drop prior to 4 hour post administration in rats in vivo. We therefore estimated that a 6-hour period would be sufficient to examine polyamine changes following one orally administered dose. The references of the kinetics of polyamine clearance from blood are now added to the revised Materials and Methods section (line 228).

References:

Rosenblum MG, Russell DH. Conjugation of radiolabeled polyamines in the rat. Cancer Res. 1977 Jan;37(1):47-51. PMID: 830421.

Okumura S, Teratani T, Fujimoto Y, Zhao X, Tsuruyama T, Masano Y, Kasahara N, Iida T, Yagi S, Uemura T, Kaido T, Uemoto S. Oral administration of polyamines ameliorates liver ischemia/reperfusion injury and promotes liver regeneration in rats. Liver Transpl. 2016 Sep;22(9):1231-44. doi: 10.1002/lt.24471. Epub 2016 Jul 22. PMID: 27102080.

5. Is there any direct evidence that ODC activity was suppressed?

Authors’ response:

ODC can be regulated by polyamines through different mechanisms, one of which is Antizyme 1-dependent prevention of ODC holoenzyme formation and another is prevention of its synthesis at the ribosomal level. In this study, we did not measure ODC activity or levels for two reasons: 1) ODC activity would be relevant only for explaining the reduction in blood levels of putrescine with long term supplementation, which is not the most abundant or bioactive polyamine in the context of the known benefits of higher polyamines, spermidine and spermine, and 2) putrescine levels did not change in animal tissues in response to 28-day supplementation. We therefore included the discussion of possible mechanisms explaining a decrease in blood putrescine levels (modulation of ODC function), but attempted to focus this study on the elevation of spermidine and spermine, which are implicated in promoting better health and longevity.

Reviewer 2 comments

1. In section 3.2, the authors mentioned the presence of liposomal nanoparticles (NPs) in spermidine-rich SRS, TCF and TNF supplements. However, there is no evidence of toxicity measurement or analysis shown by the authors. The author should also consider detecting other NPs.

Authors’ response:

We appreciate the Reviewer’s questions regarding the presence of nanoparticles in tested supplements. Because the nanoparticles we detected were not created in vitro, as done in drug delivery approaches, but assemble naturally during the extraction process of animal-derived tissues (these nanoparticles are absent from freeze dried bovine thymus tissue that was not subject to extraction procedures), we did not conduct in vivo rat toxicity measurements with these nutritional supplements (SRS, TCF and TNF), which are also commercially available for human consumption for several decades. Our published work suggests that oral administration of these supplements at the doses used in the current study improves the functional recovery of rats from traumatic brain injury, including cognitive function and the molecular biomarkers of tissue and organ recovery. However, our ongoing human clinical safety study of the TNF supplement, including its escalating doses, will reveal potential health concerns (none were detected at 6 weeks of supplementation).

Reference:

Surzenko N, Bastidas J, Reid RW, Curaba J, Zhang W, Bostan H, Wilson M, Dominique A, Roberson J, Ignacio G, Komarnytsky S, Sanders A, Lambirth K, Brouwer CR, El-Khodor BF. Functional recovery following traumatic brain injury in rats is enhanced by oral supplementation with bovine thymus extract. FASEB J. 2024 Feb 15;38(3):e23460. doi: 10.1096/fj.202301859R. PMID: 38315443.

2. The authors should also provide an illustrative model on a potential mechanism of how nuclear fraction of bovine thymus supplement can increase spermidine levels.

We thank the Reviewer for the suggestion to add a schematic model of how TNF may increase blood spermidine levels. Unfortunately, we do not have enough detail to suggest such a model. As mentioned earlier, TNF is a complex matrix of animal tissue that harbors many bioactive components and structures. Besides polyamines, TNF is likely to contain proteins, fats, RNA and many other soluble compounds, which all may contribute to the bioavailability of a given compound. We therefore would like to refrain from suggesting a model or a mechanism whereby TNF’s benefits manifest until more research is conducted on this particular supplement.

3. Also, the authors should mention how the current described strategy will provide a way to decrease age related diseases.

Authors’ response:

We appreciate the Reviewer’s suggestion to describe the way thymus extract supplementation can help decrease age-related diseases. We have verified that we included relevant information on the importance of polyamines for multiple processes that are involved in the development and progression of age-related diseases in the Discussion section of the manuscript (lines 588-600) and in the Introduction (lines 44-58). Because this study tested existing nutritional supplements and not necessarily a newly developed strategy, we attempted to focus the Discussion on the relevance of increasing the dietary supply of bioavailable polyamines through supplementation with TNF to diverse physiological processes involved in disease. The novelty of this study lies in the facts that 1) successful and clinically significant elevation of blood and tissue levels of polyamines through supplementation is yet to be demonstrated, and 2) this is the first demonstration that animal tissue extracts could be the sources of bioavailable polyamines.

4. In animal models the rat ages and relevant details need to be mentioned.

Authors’ response:

We thank the Reviewer for this suggestion and have revised the Materials and Methods section of the manuscript to include the ages of the Sprague-Dawley rats used in this study (lines 208-210). We verified that sufficient details relevant to animal housing are included in this section of Materials and Methods.

---

## [Decision Letter · Decision Letter 1]

22 Aug 2025

Nutritional supplement containing a nuclear fraction of bovine thymus gland increases the circulating levels of spermidine

PONE-D-25-09540R1

Dear Dr. El-Khodor,

We’re pleased to inform you that your manuscript has been judged scientifically suitable for publication and will be formally accepted for publication once it meets all outstanding technical requirements.

Kind regards,

Ajit Prakash, PhD

Academic Editor

PLOS ONE

Additional Editor Comments (optional):

Reviewers' comments:

Reviewer's Responses to Questions

**Comments to the Author**

1. If the authors have adequately addressed your comments raised in a previous round of review and you feel that this manuscript is now acceptable for publication, you may indicate that here to bypass the “Comments to the Author” section, enter your conflict of interest statement in the “Confidential to Editor” section, and submit your "Accept" recommendation.

Reviewer #1: (No Response)

Reviewer #2: All comments have been addressed

2. Is the manuscript technically sound, and do the data support the conclusions?

Reviewer #1: Yes

Reviewer #2: Yes

3. Has the statistical analysis been performed appropriately and rigorously? 

Reviewer #1: Yes

Reviewer #2: Yes

4. Have the authors made all data underlying the findings in their manuscript fully available?

Reviewer #1: Yes

Reviewer #2: Yes

5. Is the manuscript presented in an intelligible fashion and written in standard English?

Reviewer #1: Yes

Reviewer #2: Yes

6. Review Comments to the Author

Reviewer #1: (No Response)

Reviewer #2: The responses to comments submitted by the authors are valid. Therefore, the manuscript is ready for publication.

7. PLOS authors have the option to publish the peer review history of their article (what does this mean? ). If published, this will include your full peer review and any attached files.

**Do you want your identity to be public for this peer review?** For information about this choice, including consent withdrawal, please see our Privacy Policy .

Reviewer #1: **Yes: ** Harpreet Kaur

Reviewer #2: No

---

## [Editor Report · Acceptance letter]

PONE-D-25-09540R1

PLOS ONE

Dear Dr. El-Khodor,

I'm pleased to inform you that your manuscript has been deemed suitable for publication in PLOS ONE. Congratulations! Your manuscript is now being handed over to our production team.

Kind regards,

on behalf of

Dr. Ajit Prakash

Academic Editor

PLOS ONE